# Exercise Training Impacts Cardiac Mitochondrial Proteome Remodeling in Murine Urothelial Carcinoma

**DOI:** 10.3390/ijms20010127

**Published:** 2018-12-31

**Authors:** Rita Ferreira, Maria João Neuparth, Rita Nogueira-Ferreira, Sandra Magalhães, Miguel Aroso, José António Bovolini, Lúcio Lara Santos, Paula Oliveira, Rui Vitorino, Daniel Moreira-Gonçalves

**Affiliations:** 1QOPNA-Química Orgânica, Produtos Naturais e Agroalimentares, Departamento de Química, Universidade de Aveiro, Campus Universitário de Santiago, 3810-193 Aveiro, Portugal; ritaferreira@ua.pt; 2CIAFEL-Centro de Investigação em Actividade Física, Saúde e Lazer, Faculdade de Desporto, Universidade do Porto, R. Dr. Plácido da Costa 91, 4200-450 Porto, Portugal; mneuparth@hotmail.com (M.J.N.); jabovolini@hotmail.com (J.A.B.); 3iBiMED-Institute of Biomedicine, Departamento de Ciências Médicas, Universidade de Aveiro, Agra do Crasto, 3810-193 Aveiro, Portugal; sandra.vicencia@ua.pt (S.M.); rvitorino@ua.pt (R.V.); 4Departamento de Cirurgia e Fisiologia, Faculdade de Medicina, Universidade do Porto, Alameda Professor Hernâni Monteiro, 4200-319 Porto, Portugal; 5i3S-Instituto de Investigação e Inovação em Saúde, Universidade do Porto, 4020-135 Porto, Portugal; miguel.aroso@i3s.up.pt; 6INEB-Instituto de Engenharia Biomédica, Universidade do Porto, Rua Alfredo Allen, 208, 4200-180 Porto, Portugal; 7Grupo de Patologia e Terapêutica Experimental, Instituto Português de Oncologia, R. Dr. António Bernardino de Almeida, 4200-072 Porto, Portugal; llarasantos@gmail.com; 8CITAB-Centre for the Research and Technology of Agro-Environmental and Biological Sciences, Departamento de Ciências Veterinárias, Universidade de Trás-os-Montes e Alto Douro, Quinta de Prados, 5001-911 Vila Real, Portugal; pamo@utad.pt

**Keywords:** cardiac remodeling, treadmill running, HPLC–MS/MS, cardiac morphogenesis, metabolism

## Abstract

Cardiac dysfunction secondary to cancer may exert a negative impact in patients’ tolerance to therapeutics, quality of life, and survival. The aim of this study was to evaluate the potential therapeutic effect of exercise training on the heart in the setting of cancer, after diagnosis. Thus, the molecular pathways harbored in heart mitochondria from a murine model of chemically-induced urothelial carcinoma submitted to 8-weeks of high intensity treadmill exercise were characterized using mass spectrometry-based proteomics. Data highlight the protective effects of high intensity exercise training in preventing left ventricle diastolic dysfunction, fibrosis, and structural derangement observed in tumor-bearing mice. At the mitochondrial level, exercise training counteracted the lower ability to produce ATP observed in the heart of animals with urothelial carcinoma and induced the up-regulation of fatty acid oxidation and down-regulation of the biological process “cardiac morphogenesis”. Taken together, our data support the prescription of exercise training after cancer diagnosis for the management of disease-related cardiac dysfunction.

## 1. Introduction

Cardiac dysfunction is a recognized outcome of cancer-induced cachexia, with alterations in heart function leading to poor outcomes such as dyspnoea, fatigue, reduced quality of life, limited tolerance to therapeutics, and survival [1,2]. Until recently, cardiac function in cancer patients was a matter of concern because of the adverse effects of anticancer therapy that relies on many different classes of cardiotoxic drugs [3]. However, it is becoming more evident that cardiac function can be impaired by tumor burden per se and not only by therapeutic drugs [4,5,6]. Malignant cells were reported to produce vasoactive peptides (such as vasopressin or endothelin-1) [7,8], as well as cardiac hormones (e.g., atrial natriuretic peptide and B-type natriuretic peptide) [8,9] that affect cardiac functionality. Higher levels of troponin T were also detected in treatment-naïve cancer patients [10]. All these markers are strongly related to mortality, implying a direct association with disease progression [8].

Epidemiological evidence supports exercise as a pragmatic countermeasure to protect against cardiac injury, particularly endurance exercise. Indeed, exercise training has a great capacity to modulate numerous physiological processes in the organism, and offers a great adjuvant therapeutic strategy for the management of cardiovascular diseases [11]. Enhancement of cardiac contractile function, angiogenesis, changes in fibrillary collagen content and organization, enhanced neuronal nitric oxide synthase signaling within myocytes, activation of the IGF-1/PI3K/Akt/mTOR pathway, and mitochondrial improvements are among some of the cardioprotective changes promoted by exercise training (reviewed by ref. [12]). Regarding its impact on modulating cardiac changes secondary to cancer-cachexia, the evidence is scarce and limited to pre-clinical studies as recently reviewed [13]. Briefly, 35 weeks of endurance training prevented the mammary tumorigenesis-related increase of serum and cardiac TWEAK and prevented maladaptive cardiac remodeling [6]. Regarding its therapeutic effects, 13 weeks of moderate endurance training (started after cancer diagnosis) were sufficient to reduce cardiac fibrosis and cardiomyocyte atrophy, while significantly increasing the expression of proteins related to anti-oxidant, regeneration and mitochondrial biogenesis processes [14]. Further clarifying the mechanisms of the beneficial effects of exercise training in the cancer setting may lead to the identification of therapeutic targets.

The goal of the present study was to evaluate the effect of exercise training on cardiac function and on the molecular pathways harbored in mitochondria, following the establishment of urothelial carcinoma. Focus was given to mitochondria since the heart is highly reliant on this organelle for ATP production to support cardiac functioning [12,15]. For this purpose, we used a murine model of chemically-induced urothelial carcinoma (obtained by the administration of *N*-butyl-*N*-(4-hydroxybutyl) nitrosamine (BBN)) submitted to 8-weeks of high intensity treadmill exercise. We hypothesized that tumor-related cardiac dysfunction in mice is modulated by high intensity exercise training through the regulation of mitochondrial proteome plasticity in the heart.

## 2. Results

### 2.1. Characterization of Mice Response to BBN Exposure and/or Endurance Training

Mice from BBN+Ex group presented lighter urinary bladders and the macroscopic dimension of invasive tumors was smaller in comparison to BBN+Sed. The incidence of urothelial lesions observed in exercised and sedentary mice exposed to BBN was also different: the incidence of invasive carcinoma was 5% lower in BBN+Ex and the incidence of preneoplastic lesions was also lower in BBN+Ex group (10% lower for simple hyperplasia and 25% lower for dysplasia) compared to BBN+Sed. None of the animals from CONT groups showed urinary bladder lesions (data not shown). 

No signs of cachexia were evident throughout the entire experimental protocol, as corroborated by the absence of significant variations of body weight or heart mass after 12-weeks of BBN exposure in the drinking water followed by 8-weeks of water consumption. High intensity exercise training had no impact on body and heart weight (Table 1).

These results were related to the absence of a significant inflammatory response induced by the tumor. Indeed, no significant variations of serum levels of serum C-reactive protein (CRP) or in the pro-inflammatory cytokine TWEAK were noticed in BBN mice (Figure 1). Nevertheless, exposure to BBN resulted in a significant decrease of irisin levels (*p* < 0.05 vs. CONT groups; Figure 1), not counteracted by exercise training. Decreased serum levels of this myokine, which has been suggested to have a cardioprotective role [16], might be related to a diminishing in energy expenditure in mice.

### 2.2. Characterization of Cardiac Function and Morphometry

Baseline hemodynamic evaluation revealed left ventricular diastolic dysfunction in BBN+Sed mice and exercise training prevented this change (Figure 2, Tau). Under hemodynamic stress (induced by sudden and transient acute ascending aortic pressure overload), the perturbations of diastolic function were further exacerbated in BBN+Sed mice. In the face of the same magnitude of acute pressure overload, BBN+Ex group also responded with a significant prolongation of relaxation, but this change was significantly less in comparison to BBN+Sed (Figure 2, Tau_control_/Tau_test_).

At the morphological level, mice from BBN+Sed group presented structural derangement of some cardiomyocytes, which was less evident in BBN+Ex. Regarding fibrosis, we also qualitatively detected more fibrosis in BBN+Sed animals, with exercise training preventing its increase in BBN+Sed (Figure 3).

### 2.3. Proteomic Profiling of Cardiac Mitochondria

Since the heart is highly reliant on mitochondria for ATP production to support cardiac function, we focused our study on this organelle. We verified that tumor-bearing mice presented lower functional mitochondria as suggested by the lower ATP synthase and citrate synthase activities and OXPHOS subunits levels. Indeed, a lower ability to produce ATP was observed in BBN mitochondria (*p* < 0.05 vs. CONT groups; Figure 4), which was related to decreased content of OXPHOS complex I subunit NDUFB8 (*p* < 0.001 vs. CONT groups) and complex II subunit 30 kDa (*p* < 0.001 vs. CONT groups). Curiously, in sedentary animals, higher levels of complex III core protein 2 subunit (*p* < 0.001 vs. CONT+Sed), complex IV subunit 1 (*p* < 0.01 vs. CONT+Sed) and complex V subunit alpha (*p* < 0.001 vs. CONT+Sed) were observed. High intensity exercise training promoted the overexpression of OXPHOS complexes subunits, more notorious in CONT mice (*p* < 0.001 vs. CONT+Sed; Figure 4).

In BBN mice, high intensity exercise training promoted a significant increase of OXPHOS complexes I, II, and V subunits content (Figure 4). This mitochondrial adaptation to urothelial carcinoma and/or exercise training was better explored using a proteomic approach based on protein profiling. GeLC–MS/MS analysis of mitochondria isolated from cardiac muscle allowed the identification of 373 distinct proteins (confidence level higher than 95%; Appendix A), most of which belong to “generation of precursor metabolites and energy”, “muscle contraction” and “cellular respiration”. More than 50% of the proteins are assigned to mitochondria, as reported at MitoCarta (www.broadinstitute.org/mouse.mitocarta.2.0.html). A significant proportion of the identified proteins had more than one subcellular location, similarly to that previously observed [15]. From the identified proteins, 155 proteins were common to all experimental groups (Appendix A). Label-free quantitative analysis of these common proteins highlighted the effect of BBN exposure on the up-regulation of OXPHOS subunits as, for instance NADH dehydrogenase 1 beta subcomplex subunit 6 (NDUB6), and of the tripartite motif-containing protein 72 (TRIM72), a key component of the sarcolemmal membrane-repair machinery involved in the maintenance of cell integrity [17]. Mitochondria from BBN mice showed significant lower levels of the metabolic proteins involved in the oxidation of short fatty acids such as short-chain specific acyl-CoA dehydrogenase (ACADS), hydroxyacyl-coenzyme A dehydrogenase (HCDH), and of the metabolic protein dihydrolipoyllysine-residue acetyltransferase component of pyruvate dehydrogenase complex (ODP2), and the transport protein ADP/ATP translocase 1 (ADT1) (Figure 5). In BBN mice, high-intensity exercise training induced the overexpression of OXPHOS subunits such as cytochrome c oxidase subunit NDUFA4 (NDUA4), NADH dehydrogenase (ubiquinone) 1 beta subcomplex subunit 5 (NDUB5), cytochrome c oxidase subunit 6B1 (CXUB6) and cytochrome b-c1 complex subunit 6 (QCR6), and the down-regulation of ATP synthase subunit f (ATPK) and pyruvate dehydrogenase E1 component subunit alpha (ODPA) (Figure 5).

The analysis of the unique cardiac mitochondrial proteins per experimental group and the ones in distinct levels (higher than 1.5) showed a predominance of the biological process “cardiac tissue morphogenesis” in BBN mice whereas in healthy animals “cellular respiration” and “negative regulation of mitochondrial membrane permeability” were the most represented ones (Figure 6). Exercise training induced the up-regulation of the biological processes “cardiac muscle contraction”, “ATP metabolism” and “Smad signaling transduction” and led to the down-regulation of “cardiac morphogenesis” (Figure 6). So, exercise training seems to counteract, at least to some extent, the effect of BBN exposure through the regulation of cardiac morphogenesis. For this biological process contributes the proteins Xin actin-binding repeat-containing protein 2 (Xirp2), troponin T (Tnnt2), myosin light chain (Myl3) (Figure 6). The association of contractile proteins to mitochondria might reflect the maximization of the contractile activity through ATP channeling. Indeed, the presence of cytosolic proteins in mitochondrial fractions was previously reported in disease states [15].

## 3. Discussion

Despite the well-known cardiac impairment induced by some anticancer treatment options, little is known about cancer-induced cardiac damage independent of anticancer therapy [8]. Our data shows that BBN-induced urothelial carcinoma is associated with hemodynamic signs of cardiac dysfunction, further supported by histological evidences of fibrosis and significantly lower serum levels of the cardioprotector myokine irisin. The spectrum of lesions observed in the urinary bladder of mice exposed to BBN is similar to the ones observed in humans [18], with cachexia occurring at advanced stages of the disease [19]. In the present study, mice did not show signs of systemic inflammation, as the circulating levels of CRP and TWEAK were not increased, and this was paralleled by the absence of muscle wasting. Thus, our data suggest that tumor burden per se induces cardiac remodeling that precedes muscle wasting, which might be exacerbated when cachexia is established, as previously reported [20,21]. This cardiac remodeling observed in tumor-bearing mice was characterized by alterations on mitochondrial proteome with emphasis on the up-regulation of the biological process “cardiac tissue morphogenesis” and the down-regulation of “cellular respiration”. 

Cellular respiration was found to be down-regulated in the heart of tumor-bearing mice (Figure 4 and Figure 6), with significant impact on the ability to produce ATP (Figure 4). The higher levels of Xirp2 were probable contributors for the up-regulation of heart morphogenesis observed in this group of animals (Figure 6). Xirp2 is a Xin protein that is responsible for mechanical and electrical communication among adjacent cardiomyocytes [22]. This family of proteins plays an important role in modulating stress responses in the adult heart, being up-regulated in early stages of exposure to cardiac stressing stimuli [23], eventually to help dealing with sarcomeric lesions [24]. However, to the best of our knowledge, the association of Xirp2 to mitochondria has never been explored. The translocation of non-mitochondrial proteins to this organelle has been reported in stress conditions [25,26]. So, BBN-related higher levels of mitochondrial Xirp2 seem to reflect cancer-induced cardiac remodeling.

Label-free proteome quantitation evidenced the down-regulation of enzymes involved in fatty acid oxidation in BBN heart as short-chain acyl-CoA dehydrogenase and hydroxyacyl-CoA dehydrogenase (Figure 5), the major source of energy supply in this organ [12,27,28]. Such metabolic remodeling towards the utilization of other metabolic substrates as carbohydrates was previously reported in conditions of heart failure [27,29]. Derangements in cardiac energy metabolism contribute to the pathological remodeling of heart once the capacity of cardiac mitochondria to generate ATP is compromised and myocardial high-energy phosphate stores, specifically phosphocreatine levels, the main reservoir sources of ATP, are reduced [29]. 

Exercise training is widely recognized to promote a unique cardioprotective phenotype, being a useful therapeutic strategy for the management of cardiovascular diseases. The exercise-induced cardioprotective phenotype is explained, at least in part, by important alterations in the mitochondrial proteome [30]. In the cancer setting, there are a limited number of studies exploring the cardioprotective role of exercise training against cancer-induced cardiac remodeling [5] and none have explored the modifications occurring at the level of cardiac mitochondria. Herein, we verified that high-intensity exercise training counteracts BBN-induced up-regulation of the biological process “cardiac morphogenesis”, and promotes the overexpression of proteins belonging to the biological processes “cardiac muscle contraction” and “ATP metabolism”, which altogether could support the improved hemodynamic response observed in trained mice (Figure 2). Indeed, high intensity exercise training counteracted the metabolic remodeling observed in the heart of sedentary BBN mice by promoting the up-regulation of fatty acid metabolism, evidenced by the higher content of long-chain specific acyl-CoA dehydrogenase (Acadl), ATP metabolism and, consequently, muscle contraction (Figure 6). So, our data support the efficient glucose and fatty acid handling promoted by chronic exercise training in the heart of tumor-bearing mice, similarly to the previously reported for healthy subjects [27]. Because exercise-induced adaptations are dependent on the type (e.g., endurance versus resistance versus combined exercise) and volume (e.g., frequency, duration and intensity of the exercise bout), further work is needed to clarify which is the best protocol to maximize protection against cancer-induced cardiac remodeling [13]. Moreover, it is known that vigorous exercise imposes a greater relative risk of sudden cardiac death in those individuals with underlying cardiac disease compared with those during rest [31]. Despite no deaths occurring in our exercising rats with cancer, the safety of high intensity exercise should be addressed in future clinical studies. Finally, our rats did not develop cachexia throughout the protocol. Because the safety and effectiveness of exercise in cachexia is not yet established, high intensity running exercise might not be recommended in advanced stages of cancer.

## 4. Materials and Methods 

### 4.1. Animals and Experimental Design

The animal protocol was approved by the Portuguese Ethics Committee for Animal Experimentation, *Direção Geral de Alimentação e Veterinária* (licence number 020157; 2014-09-24) and was performed in accordance to European Commission Recommendation 2007/526/CE. Forty four ICR (CD1) male mice were obtained at the age of 4 weeks from Harlan (Barcelona, Spain). After a week’s quarantine, mice were handled daily to get them accustomed to human touch; they were placed under controlled conditions of 23 ± 2 °C, 50 ± 10% humidity, and 12 h light/dark cycle. Animals were maintained on a standard diet during all the experiments. Mice were then randomly divided into two groups: exposed to 0.05% *N*-butyl-*N*-(4-hydroxybutyl) nitrosamine (BBN) in the drinking water over the course of 12 weeks (BBN group, *n* = 30) and with access to tap water (CONT group, *n* = 14). According to previous experience with this animal model, tumor burden is expected after 12 weeks of oral exposition to BBN [32]. At this point, half of the animals from each group started an exercise program in a treadmill running over 8 weeks (subgroups BBN+Ex (*n* = 15) and CONT+Ex (*n* = 7)). The number of animals per group was planned taking into consideration the potential biological variability to BBN exposure and to exercise training, mortality rate and 3Rs (Replacement, Reduction and Refinement) policy of animal experimentation [33]. All the animals concluded the protocol.

Mice from Ex groups were submitted to high intensity training (HIT) for 8 weeks, 5 days/week. HIT consisted of 10 bouts of 4 min high intensity running at 25 m/min, interspersed by 2 min of active rest at 10 m/min, with the treadmill at an inclination of 10°. The interval pace was increased gradually from 10 to 25 m/min over the course of the first 4 weeks and maintained at this value for the rest of the exercising period. The average distance covered by session was 1031 meters. Before the first interval, each mouse performed a regular warm-up of 10 min at a speed of 10 m/min. From the 15 mice of BBN+Ex group, 14 successfully accomplished this exercise program. Mice from sedentary (Sed) group were familiarized with the treadmill by performing 10 minutes of continuous running on the treadmill 3 times per week at a running speed of 10 m/min, with 10° inclination.

At the end of the protocol, all animals were submitted to a treadmill maximal exercise running test slightly modified from that described by Kemi et al. [34]. After 10 min of warm-up at a speed of 10 m/min and 10° inclination, the speed was progressively increased 2 m/min every 2 min until exhaustion. Exhaustion was considered when animals were unable or refused to run further for 3 consecutive times despite stimulation. Mice that performed exercise training showed greater exercise performance in comparison to their respective sedentary counterparts (Appendix A). Forty-eight hours after the last training session, hemodynamic evaluation was performed. 

### 4.2. Hemodynamic Evaluation

Animals were anesthetized by inhalation of a mixture of sevoflurane (8% for induction and 2.5%–3% for maintenance) and oxygen, endotracheally intubated for mechanical ventilation (150 min^−1^, 100% O_2_, 14–16 cm H_2_O inspiratory pressure, with tidal volume adjusted to animal weight, and 5 cm H_2_O end-expiratory pressure; TOPO Small Animal Ventilator—Kent Scientific, Dual Mode, San Francisco, CA, USA) and placed over a heating pad. Under binocular surgical microscopy (Leica, Wild 384000, Wetzlar, Germany), the right jugular vein was cannulated for fluid administration (prewarmed 0.9% NaCl solution, 32 mL/kg/h) to compensate for perioperative losses. The heart was exposed through a median sternotomy. A catheter was placed in the left ventricle in order to obtain hemodynamic data (PVR-1045, Millar Instruments, Houston, TX, USA). After complete instrumentation, the animal preparation was allowed to stabilize for 15 min. Hemodynamic recordings were made with respiration suspended at end-expiration under basal conditions and during acute pressure overload (sudden and transient occlusion of ascending aorta) as previously described by us [35]. Data were continuously acquired (MPVS 300, Millar Instruments, Houston, TX, USA), digitally recorded at 1000 Hz (ML880 PowerLab 16/30, Millar TM Instruments), and analyzed (LabChart, ADInstruments). After complete hemodynamic assessment, animals were euthanized by exsanguination through cardiac puncture as indicated by the Federation for Laboratory Animal Science Association [36] under anesthesia. Blood, urinary bladder, and cardiac muscle samples were collected. The urinary bladders were inflated in situ by intravesical instillation of 100 microliter of buffered 10% phosphate formalin solution for 12 h. Urinary bladders were processed for histology. Cardiac muscle was weighed and divided for morphological and biochemical analysis.

### 4.3. Blood Tests

Serum C-reactive protein (CRP), TWEAK, and irisin levels were assayed by immunoblotting. Serum was diluted in Tris buffered saline (TBS) to obtain a final protein concentration of 0.001 μg/μL and a volume of 100 μL was slot-blotted into a nitrocellulose membrane (Whatman^®^, Protran^®^, Dassel, Germany). Immunodetection was performed as described below.

### 4.4. Histological Analysis of Urinary Bladder and Cardiac Muscle

Cubic pieces from cardiac muscle and urinary bladder were fixed (4% (*v*/*v*) buffered paraformaldehyde) by diffusion during 24 h and subsequently dehydrated with graded ethanol and included in paraffin blocks. Serial sections (5 µm of thickness) of paraffin blocks were cut by a microtome and mounted on silane coated slides. The slides were dewaxed in xylene and hydrated through graded alcohol finishing in phosphate buffered saline solution. Deparaffinized sections of cardiac tissue were stained with hematoxylin-eosin or Picrosirius red. Due to constraints regarding the amount of cardiac sample, it was not possible to perform a quantitative histological analysis. Urinary bladder histology was performed to confirm the spectrum of urothelial lesions previously observed [37].

### 4.5. Mitochondria Isolation

Heart tissue was used for the preparation of isolated mitochondria, as previously described [15,38]. All the procedures were performed on ice or below 4 °C. Briefly, heart sections from two mice were pooled and then minced in an ice-cold isolation medium containing 250 mM sucrose, 0.5 mM EGTA, 10 mM HEPES-KOH (pH 7.4) and 0.1% defatted bovine serum albumin (BSA, Sigma, Saint Louis, MO, USA). The minced blood-free tissue was resuspended in isolation medium containing protease subtilopeptidase A type VIII (1 mg/g tissue) and homogenized with tightly fitted Potter–Elvehjen homogenizer and Teflon pestle. The suspension was incubated for 1 min (4 °C), re-homogenized and centrifuged at 14,500× *g* over 10 min. The supernatant fluid was decanted, and the pellet, essentially devoid of protease, was gently resuspended in isolation medium. The suspension was centrifuged (750× *g*, 10 min) and the resulting supernatant was centrifuged again (12,000× *g*, 10 min). The new pellet was resuspended and re-pelleted (12,000× *g*, 10 min). Finally, the pellet, containing the mitochondrial fraction, was gently resuspended in a washing medium containing 250 mM sucrose, 10 mM HEPES-KOH, pH 7.4. Phosphatase and protease inhibitors (P2850, P5726, P8340; Sigma, Saint Louis, MO, USA) were added and all the procedures were performed at 4 °C. Protein concentration was estimated with a colorimetric method (RC–DC protein assay, Bio-Rad, Hercules, CA, USA) using BSA as standard.

### 4.6. Determination of ATP Synthase Activity

ATP synthase activity was measured in mitochondrial fractions as previously described [39]. The phosphate produced by hydrolysis of ATP reacts with ammonium molybdate in the presence of reducing agents to form a blue-color complex, the intensity of which is proportional to the concentration of phosphate in solution. Oligomycin was used as an inhibitor of mitochondrial ATPase activity.

### 4.7. Determination of Citrate Synthase (CS) Activity

CS activity was measured according to Coore et al. [40]. In brief, the CoASH released from the reaction of acetyl-CoA with oxaloacetate was measured by its reaction with 5,5’-dithiobis-(2-nitrobenzoic acid) (DTNB) at 412 nm (molar extinction coefficient of 13.6 mM^−1^ cm^−1^).

### 4.8. Immunoblotting Analysis

Equivalent amounts of mitochondrial protein were electrophoresed on a 12.5% SDS-PAGE as described by Laemmli [41]. Gels were blotted onto a nitrocellulose membrane (Whatman^®^, Protran) in transfer buffer during 2 h (200 mA). In these membranes and in the ones obtained by slot-blot of serum samples, nonspecific binding was blocked with 5% (*w*/*v*) dry nonfat milk in Tris buffered saline with Tween 20 (TBS-T). Then, membranes were incubated with primary antibody (mouse anti-Total OXPHOS, ab110413; rabbit anti-TWEAK, ab37170; rabbit anti-CRP, ab32412; or rabbit anti-irisin, ab174833, from Abcam) for 2 h at room temperature, washed and incubated with secondary with horseradish peroxidase-conjugated anti-mouse or anti-rabbit (GE Healthcare). Immunoreactive bands were detected by enhanced chemiluminescence ECL (GE Healthcare) according to the manufacturer’s procedure and images were recorded using X-ray films (Kodak Biomax Light Film, Sigma). Films were scanned in Molecular Imager Gel Doc XR+System (Bio-Rad) and analyzed with Image Lab software (v4.1, Bio-Rad). Protein loading was controlled by Ponceau S staining since no ideal protein marker is known for mitochondria.

### 4.9. GeLC-MS/MS Analysis of Isolated Mitochondria

Protein extract corresponding to 40 μg of mitochondrial extracts (*n* = 3 for each experimental group) was separated using SDS-PAGE in a 12.5% gel prepared as previously described [41]. The gel was stained with Colloidal Coomassie Blue G250. Then, complete lanes were cut out of the gel and sliced into 16 sections. Each section was in-gel digested with trypsin (Pierce, Life Technologies, Rockford, IL, USA). The resulting peptide mixture was then extracted from the gel fractions and dried using vacuum centrifugation. The dried extracted peptides were dissolved in 5 μL of mobile phase A (0.1% trifluoroacetic acid (TFA), 5% acetonitrile (ACN), 95% water). Tryptic digests were separated using a 150 mm × 75 µm Pepmap100 capillary analytical C18 column with 3 µm particle size installed in an Ultimate 3000 (Dionex, Sunnyvale, CA, USA) at a flow rate of 300 nL/min. The gradient started at 10 min and ramped to 50% Buffer B (85% ACN, 0.045 % TFA) over a period of 45 min. The peptides eluting from the column were mixed with a continuous flow of matrix solution (270 nL/min, 2 mg/mL α-CHCA in 70% ACN/0.3% TFA and internal standard Glu-Fib at 15 fmol) in a fraction microcollector (Probot, Dionex/LC Packings, Sunnyvale, CA, USA) and directly deposited onto the LC-MALDI plates at 22 second intervals for each spot. Samples were then analyzed using a 4800 MALDI-TOF/TOF Analyzer (AbSCIEX, Foster City, CA, USA). A S/N threshold of 50 was used to select peaks for MS/MS analyses. The spectra were processed using the TS2Mascot (v1.0, Matrix Science Ltd., London, UK) and submitted to Mascot software (v.2.1.0.4, Matrix Science Ltd.) for peptide/protein identification. Searches were performed against the SwissProt protein database (March 2016) for *Mus musculus*. Search was performed including data from all slices for global protein identification and emPAI calculation. An MS tolerance of 30 ppm was found for precursor ions and 0.3 Da for fragment ions, as well as two missed cleavages, methionine oxidation and propionamide as variable modification and carbamidomethyl as fixed modification. An independent False Discovery Rate (FDR) analysis using the target-decoy approach provided with Mascot software was used to assess the quality of the identifications and positive identifications were considered when identified proteins and peptides reached a 5% local FDR. Furthermore, proteins identified with one peptide were manually validated when MS/MS spectra presented at least 4 successive amino acids covered by *b* or *y* fragmentations. For label-free quantification and abundance estimation, common identified proteins to all groups were considered for analysis by calculating the exponentially modified protein abundance index (emPAI) [42]. The emPAI is an exponential form of PAI^−1^ (the number of detected peptides divided by the number of observable peptides per protein, normalized by the theoretical number of peptides expected via in silico digestion) defined as emPAI = 10^PAI^ − 1 and the corresponding protein content in mole percent is calculated as mol% = (emPAI/∑emPAI) × 100. Microsoft Office Excel was used to calculate the mole percent. The theoretically observable peptides were determined by the in silico digestion of mature proteins using from the output of the program Protein Digestion Simulator (http://panomics.pnnl.gov/software/). The observed peptides were unique parent ions including those with two missed cleavage. Mean protein emPAI values were log2 transformed for protein ratio calculation. Proteins with fold changes higher than 1.5 were considered for bioinformatics analysis.

### 4.10. Statistical Analysis

Values are given as mean ± standard deviation for all variables. The Kolmogorov–Smirnov test was performed to check the normality of the data. The statistical significance of the differences between the experimental groups was determined using a one-way analysis of variance followed by the Tukey multiple comparisons post hoc test. Results were considered significantly different when *p* < 0.05. Statistical analysis was performed with GraphPad Prism software (version 5.0, San Diego, CA, USA).

## 5. Conclusions

Taken together, the present work provides hemodynamic, morphological, and molecular insights on the improvements promoted by high-intensity exercise training in the heart from tumor-bearing mice after cancer diagnosis. Efficiently energetic substrate handling, improved muscle contraction, and down-regulation of cardiac morphogenesis were among the molecular alterations promoted by high intensity chronic exercise training.

## Figures and Tables

**Figure 1 ijms-20-00127-f001:**
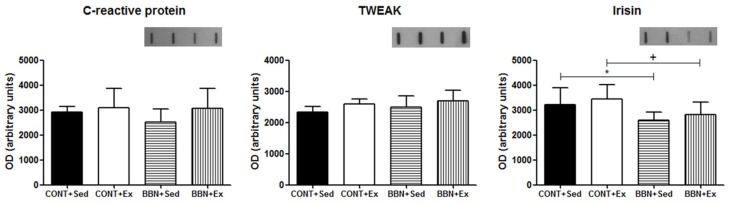
Effect of *N*-butyl-*N*-(4-hydroxybutyl) nitrosamine (BBN) exposure and/or exercise training on serum levels of C-reactive protein, TWEAK and irisin. Representative immunoblots are shown above the correspondent graph (sample order has correspondence to the order of the groups presented in the graph). Values are expressed as mean ± standard deviation of *n* = 6 (* *p* < 0.05 vs. CONT+Sed; + *p* < 0.05 vs. CONT+Ex).

**Figure 2 ijms-20-00127-f002:**
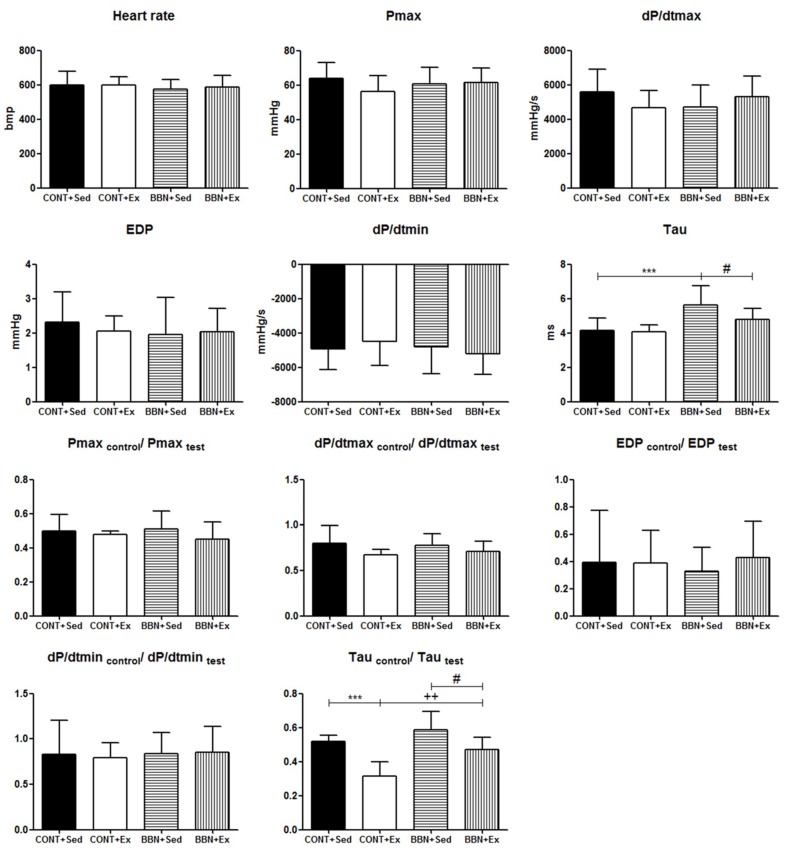
Effect of BBN exposure and/or exercise training on cardiac function in baseline and stress conditions: heart rate, left ventricular Pmax, dP/dtmax, EDP, dP/dtmin, Tau, Pmax_control_/Pmax_test_, dP/dtmax_control_/dP/dtmax_test_, EDP_control_/EDP_test_, dP/dtmin_control_/dP/dtmin_test_ and Tau_control_/Tau_test_. Pmax, maximum pressure; dP/dtmax, peak rate of pressure rise; dP/dtmin, peak rate of pressure fall; EDP, end-diastolic pressure; Tau, time constant of ventricular pressure decay. Values are expressed as mean ± standard deviation of *n* = 15 for BBN groups and *n* = 7 for CONT groups (*** *p* < 0.001 vs. CONT+Sed; ++ *p* < 0.01 vs. CONT+Ex; # *p* < 0.05 vs. BBN+Sed).

**Figure 3 ijms-20-00127-f003:**
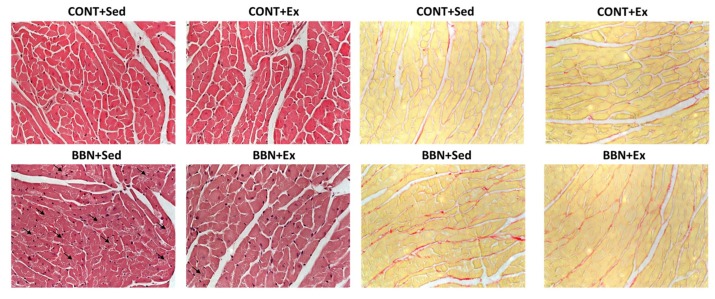
Effect of BBN exposure and/or exercise training on histological appearance of cardiomyocytes stained with hematoxylin and eosin, and on myocardial fibrosis given by Picrosirius red staining (red for collagen and yellow for cardiac muscle). All microscopic fields were obtained with a magnification of 400×.

**Figure 4 ijms-20-00127-f004:**
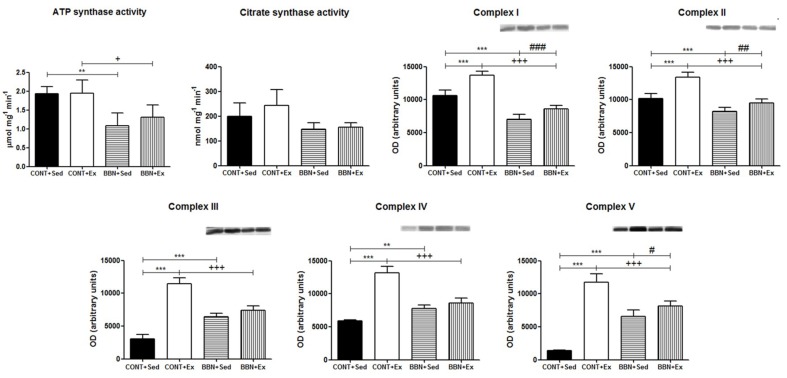
Effect of BBN exposure and/or exercise training on ATP synthase and citrate synthase activities, and on OXPHOS complexes subunits (complex I, CI-NDUFB8; complex II, CII-SDHB; complex III, CIII-UQCRC2; complex IV, CIV-MTCO1 and complex V, CV-ATP5A). Representative Western blots are shown above the correspondent graph (sample order has correspondence to the order of the groups presented in the graph). Values are expressed as mean ± standard deviation of *n* = 6 for BBN groups and *n* = 4 for CONT groups (** *p* < 0.01 vs. CONT+Sed; *** *p* < 0.001 vs. CONT+Sed; + *p* < 0.05 vs. CONT+Ex; +++ *p* < 0.001 vs. CONT+Ex; # *p* < 0.05 vs. BBN+Sed; ## *p* < 0.01 vs. BBN+Sed; ### *p* < 0.001 vs. BBN+Sed).

**Figure 5 ijms-20-00127-f005:**
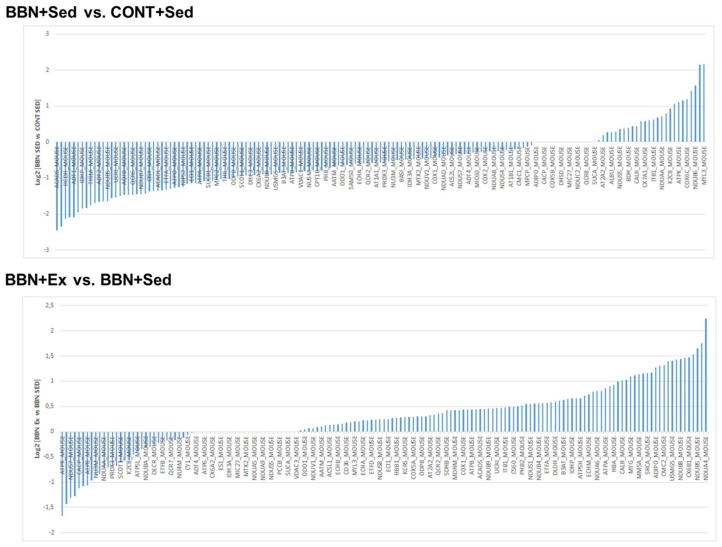
Normalized abundance of proteins in BBN+Sed vs. CONT+Sed and BBN+Ex vs. BBN+Sed groups. Proteins with log2 lower than 0.05 are not presented in the figure but are listed in Appendix A. Protein accession numbers are correlated to protein names in Appendix A.

**Figure 6 ijms-20-00127-f006:**
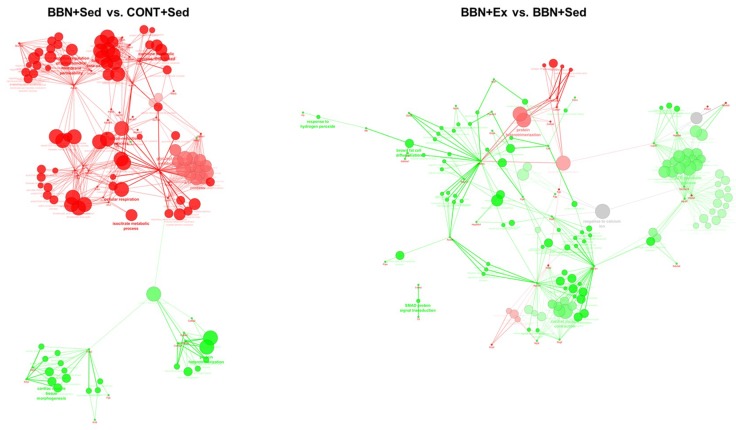
ClueGo + CluePedia analysis of protein-protein interaction considering unique proteins and the ones present in significantly distinct levels (based on emPAI values) in the heart of CONT+Sed and BBN+Sed, and BBN+Sed and BBN+Ex animals. In BBN+Sed vs. CONT+Sed analysis, green nodes refer to the biological processes up-regulated in the BBN+Sed group and red nodes refer to the ones up-regulated in the CONT+Sed group. In BBN+Ex vs. BBN+Sed analysis, green nodes refer to the biological processes up-regulated in BBN+Ex and red nodes refer to the ones up-regulated in BBN+Sed. Node size and color intensity reflect the number of associated proteins.

**Table 1 ijms-20-00127-t001:** Characterization of the animals’ response to *N*-butyl-*N*-(4-hydroxybutyl) nitrosamine (BBN) and/or exercise training regarding body weight, heart mass, muscle mass, and heart-to-body weight.

Experimental Group	Body Weight (g)	Heart Weight (g)	Gastrocnemius Mass (g)	Heart-to-Body Weight (mg/g)
CONT+Sed	39.23 ± 2.92	0.20 ± 0.03	0.40 ± 0.02	5.24 ± 0.94
CONT+Ex	39.39 ± 3.58	0.22 ± 0.04	0.35 ± 0.06	5.61 ± 0.98
BBN+Sed	40.69 ± 2.41	0.21 ± 0.03	0.42 ± 0.04	5.18 ± 0.50
BBN+Ex	40.93 ± 2.59	0.21 ± 0.04	0.39 ± 0.04	5.22 ± 1.12

Values are expressed as mean ± standard deviation of *n* = 15 for BBN groups and *n* = 7 for CONT groups.

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
