# Peer review of "Exercise Training Impacts Cardiac Mitochondrial Proteome Remodeling in Murine Urothelial Carcinoma"

_ijms, 2018, doi:10.3390/ijms20010127_

Round 1
Reviewer 1 Report
In the present paper, Authors demonstrated that exercise training induces mitochondrial adaptations, which play a critical role in protection urothelial carcinoma. It is well known that exercise training induces a cardio-protective phenotype and alterations in cardiac mitochondrial proteins. however I suggest Authors to add the citation Am J Physiol Heart Circ Physiol. 09 Jul;297(1):H144-52. doi: 10.1152/ajpheart.01278.2008. I also suggest Authors to add comment/s on the effects of the single exercise session versus the training and also specify on the effect that different protocols of exercise might induce on mitochondrial adaptations and cancer protection and which protocol can be recommended and considered adequate and safe.
The ms is clearly written and this work contributes to add new evidence on the relationship between physical activity and oncological protection and control. Data obtained support the prescription of exercise training after cancer diagnosis with particular attention on the management of disease-related cardiac dysfunction. Author hypotheses are well supported to consider the ms suitable for publication in Int. J. Mol. Sci.
Author Response
Following reviewer’s suggestion, we included the proposed paper in the discussion section of the revised version of our manuscript (page 8, line 238 ) to support the exercise-induced changes in cardiac mitochondria. Regarding the other topics highlighted by the reviewer, we added the following statement in page 8: “Because exercise-induced adaptations are dependent on the type (e.g. endurance versus resistance versus combined exercise) and volume (e.g. frequency, duration and intensity of the exercise bout), further work is needed to clarify which is the best protocol to maximize protection against cancer-induced cardiac remodelling (Trends Mol Med. 2018 Aug;24(8):709-727). Moreover, it is known that vigorous exercise imposes a greater relative risk of sudden cardiac death in those individuals with underlying cardiac disease compared with those during rest (Med Sci Sports Exerc. 2015 Nov;47(11):2473-9). Despite no deaths occurred in our exercising rats with cancer, the safety of high intensity exercise should be addressed in future clinical studies. Finally, our rats did not develop cachexia throughout the protocol. Because the safety and effectiveness of exercise in cachexia is not yet established, high intensity running exercise might not be recommended in advanced stages of cancer.”
The references to be included were added in commentary boxes due to format issues.
Reviewer 2 Report
In this study, effects of treadmill exercise on heart functions and mitochondrial proteome have been studied in control and tumour-bearing mice. Data showed that, in addition to a few changes in cardiac functions and morphology, exercise training induced expression of proteins related to oxidative phosphorylation. However, there are several points to be cosidered.
(1) Figure 5
The font size is too small to read. Any figure should be prepared so that readers can read and understand what it means. In addition, it seems that some proteins in Supplemental Table S1 are missing in Fig. 5 (ex. albu_mouse). What is the difference between Fig. 5 and Fig. S1 ?
(2) Figure 6
The text font used in this figure is so small that, it is not possible to understand what this figure mean. It was described in the Results section that several biological processes such as "cardiac muscle contraction" were up/down regulated, but readers can not realize these conclusions from this figure.
(3) Figure 6
In the Results section, it was described that proteins involved in muscle morphogenesis including troponin T and myosin light chain were upregulated by exercise training. However, since the protein samples used for the proteomic analysis were purified mitochondrial fractions, proteins other than mitochondrial ones are more likely contaminants, which cannot be used for quantitative analysis. So the Results section as well as the discussion section should be written with taking account with this.
(4) Minor points,
P9, L298 Protan -> Protran
Author Response
(1)
Fig.5 depicts the quantitative comparison of data listed in Supplemental Table S1. Following the reviewer’s concern, we improved the readability of Fig.5 by removing all the proteins whose abundance did not differ (but their identification was maintained in Supplemental Table S1). This was explained in the legend of the figure in the revised version of the manuscript. Regarding the differences between Fig. 5 and Fig. S1, while the first shows normalized abundance of proteins, figure S1 presents the effect of BBN and/or exercise on the exercise test duration, maximal speed attained, and total distance covered. We checked the supplemental figure uploaded and Figure S1 is correct.
(2)
According to the reviewer’s concern, Figure 6A and figure 6B were uploaded separately to allow the editorial office to manage figures size and thus improve its visualization by the readers.
(3)
We understand reviewer’s concern and agree that the presence of cytoplasmic proteins in mitochondrial fractions is usually a sign of contamination. However, previous studies from ours and other groups reported the presence of cytoplasmic and organelles besides mitochondria in this organelle fractions in pathophysiological conditions, which might be a sign of newly established protein-protein interactions or proteins re-allocation (Rocha et al., J Proteomics. 2011 75(1): 221-8). We hypothesize that these contractile proteins interact with mitochondria in disease in an attempt to maximize the utilization of energy for muscle contraction, contributing to muscle morphogenesis. To better clarify this issue, we introduced the following idea in the Results section: “The association of contractile proteins to mitochondria might reflect the maximization of the contractile activity through ATP channeling. Indeed, the presence of cytosolic proteins in mitochondrial fractions was previously reported in disease states (REF).”
(4)
The correction was made.
Reviewer 3 Report
In this manuscript, Ferreira et al aimed to investigate the potential therapeutic effect of exercise training on the cardiac function in the setting of cancer using a murine model. The authors focused on mitochondria pathway which is very reasonable, since myocardial function relies heavily on mitochondria, and cardiomyocytes are mitochondrial rich which allows them to produce ATP quickly. This is a very interesting and important subject for research.
Authors showed several important points:
1) Exercise can potentially decrease the tumor burden in this murine model
2) Mice with induced tumor had diastolic dysfunction compared with the control mice, and exercise can potentially improve diastolic function.
3) Exercise is associated with less myocardial fibrosis
4) Exercise is associated with enhanced mitochondrial function and more efficient ATP production
The study is well designed, and the conclusions are, in general, well supported by the data. Just one minor point: to show exercise is associated with less myocardial fibrosis, authors showed several H and E stains of cardiomyoctyes. It there quantification data available for this? Quantification analysis will be more convincing.
Author Response
We acknowledge the reviewer’s constructive comments and we agree that quantitative analysis of myocardial fibrosis would be more convincing. However, we used a mice model (and not a rat model), which limited the amount of cardiac tissue sample available for all the analysis. Because our priority was to maximize cardiac tissue for mitochondrial isolation, we could not obtain sufficient cardiac samples to perform a quantitative histological analysis. This has been highlighted in the revised version of the manuscript (page 10, line 317).